# Orientation Identification of the Black Phosphorus with Different Thickness Based on B_2g_ Mode Using a Micro-Raman Spectroscope under a Nonanalyzer Configuration

**DOI:** 10.3390/ma13235572

**Published:** 2020-12-07

**Authors:** Rubing Li, Yongchao Shang, Huadan Xing, Xiaojie Wang, Mingyuan Sun, Wei Qiu

**Affiliations:** Tianjin Key Laboratory of Modern Engineering Mechanics, Department of Mechanics, School of Mechanical Engineering, Tianjin University, Tianjin 300072, China; rubingli@tju.edu.cn (R.L.); SYCxianfeng@163.com (Y.S.); xinghuadan@163.com (H.X.); wangxiaojie2019@tju.edu.cn (X.W.); crystal119smy@outlook.com (M.S.)

**Keywords:** black phosphorus, orientation identification, B_2g_ mode, micro-Raman spectroscope, nonanalyzer configuration

## Abstract

As an anisotropic material, the unique optoelectronic properties of black phosphorus are obviously anisotropic. Therefore, non-destructive and fast identification of its crystalline orientation is an important condition for its application in optoelectronics research field. Identifying the crystalline orientation of black phosphorus through A_g_^1^ and A_g_^2^ modes under the parallel polarization has high requirements on the Raman system, while in the nonanalyzer configuration, the crystalline orientation of the thick black phosphorus may not be identified through A_g_^1^ and A_g_^2^ modes. This work proposes a new method to identify the crystalline orientation of black phosphorus of different thicknesses. This method is conducted under the nonanalyzer configuration by B_2g_ mode. The results show that B_2g_ mode has a good consistency in the identification of crystalline orientations. In this paper, a theoretical model is established to study the angle-resolved Raman results of B_2g_ mode. The new method can accurately identify the crystalline orientation with different layers of black phosphorus without misidentification.

## 1. Introduction

Phosphorene, that is black phosphorus of few layers or even monolayer [1], is one of the most attractive two-dimensional (2D) materials. Different from other 2D materials [2], including graphene [3], anisotropy is a common and distinctive feature among the acoustical [4], optical [5], thermal [6], electronical [7,8], and mechanical [9,10] properties of phosphorene, which is intensively dependent on the crystalline orientation. Black phosphorus exhibits many intriguing properties including a highly tunable bandgap [11,12], relatively high carrier mobility [1,13], high on/off ratio [14] and a negative Poisson’s ratio [15,16]. The bandgap of black phosphorus is always direct, regardless of thickness, and ranges from 0.3 to 2 eV [12,17,18], bridging the gap between zero-gap graphene and large-gap transition metal dichalcogenides [19]. Owing to the anisotropic properties, phosphorene can be prospectively applied in a new breed of nano-machine, microelectronic and optoelectronic devices [1,8,12,14,17,20,21] as a core material. Therefore, to better explore the applications and characteristics of phosphorene [22], it is crucial to achieve a precise, nondestructive and speedy recognition method of its crystalline orientation.

Regarding the crystalline orientation of 2D materials, there are several methods, such as microscopic observation using a scanning probe microscope (SPM) [23] or transmission electron microscope (TEM) [24,25], infrared or micro-Raman spectroscopy, and angle-resolved conductivity. In comparison, the contact method of angle-resolved conductivity has a lower spatial resolution and worse sensitive angle resolution [21]. It is difficult to use TEM or SPM to realize an online and nondestructive measurement. In addition, most infrared spectrographs are difficult to meet the requirement of crystalline measurement at micro-scale due to spectral and spatial resolution [8,21].

Micro-Raman spectroscopy [26] is a fast, high-resolution [27,28] and non-contact [29,30] measurement method, which has been widely used in various research fields such as optics [5] and mechanics [9,31]. Existing work discovered that the polarized micro-Raman spectroscopy under the parallel configuration can be applied to realize the identification of the crystalline orientation of phosphorene. When the polarization direction of the incident laser and scattering light are kept parallel, the crystalline orientation of the phosphorene sample can be identified from the relationship between the A_g_^2^ intensity of Raman spectrum and its corresponding polarization angle [21]. Further study proved that such an identification of crystalline orientation can also be realized under a nonanalyzer configuration, if the spectrometer does not equip the optional accessories of the parallel configuration.

However, because of the inherently complex interactions between electron, photon and phonon [5], the laser wavelength [5,32,33], sample thickness (layer numbers) [5,32,33] and strain state [9,10,34] influence the Raman intensity of different crystalline orientations, leading to nondeterminacy and even misidentification when using existing polarized Raman methods on the basis of A_g_^2^ mode to realize the identification of crystalline orientations of black phosphorus.

A new method is proposed in this paper to recognize the crystalline directions of black phosphorus. This method is based on the angle-resolved Raman detection and the theoretical model of the intensity–orientation relationship of the B_2g_ mode under the nonanalyzer configuration. In-situ measurement is achieved by changing the polarization direction of the incident laser. The crystalline orientation of black phosphorus of different thicknesses can be easily and effectively identified. In addition, the new method also reduces the requirement of Raman instrument conditions, and can realize crystalline direction identification with any Raman instrument.

## 2. Samples and Experiments

### 2.1. Samples

The phosphorene sample was mechanically peeled from black phosphorus crystalline using 3M Scotch tape. Then, the obtained sample was transferred from the tapes to a Si substrate coated with a 300 nm SiO_2_ film on the Si surface. To reduce the effect of the oxidation of black phosphorus in the air, the characterization experiment was performed immediately after obtaining the sample.

### 2.2. Experiments

This work used a Renishaw InVia Reflex confocal micro-Raman spectroscopy system (Wotton-under-Edge, UK) with a Leica 100× Lens (numerical aperture = 0.85, Wetzlar, Germany), a 1200 1/mm grating, a 633 nm laser (1.96 eV). The exposure time of each Raman detection was 4 s and the resolution of the spectroscope was 2 cm^−1^. The laser power reached the sample surface was about 0.64 mW, preventing the phosphorene from being burned out. The Leica optical microscope in the Raman system was used to observe the samples in the experiment. The experimental data were processed with origin software.

In order to compare the method proposed in this work with the existing method, two methods were used to detect the Raman data on each sampling spot in-situ. Figure 1 shows the optical diagram of polarized micro-Raman spectroscope used in this work. A 360° continuously adjustable half-wave plate (synergistic/non-synergistic continuous polarization accessory, patent CN102426163A) was inserted into the common path of incident laser and scattering light. The polarization direction of the incident laser arrived to the sample was controlled by changing the optical axis direction of the half-wave plate. If the analyzer was inserted into the scattering light path, the polarization direction of the scattering light finally arrived on the spectrograph was selected. When the optical axis of the analyzer paralleled the polarization direction of the incident laser before the half-wave plate, the polarization direction of the scattering light finally arrived on the spectrometer was always similar with that of the incident laser arrived to the sample, which was, namely, the configuration of parallel polarization (PP). If the analyzer was not inserted into the scattering light path, the Raman system is equivalent to under the nonanalyzer configuration (NA). By switching the analyzer into and out of the scattering light path, this work realized the Raman detection in-situ under both the PP and the NA configurations. During the experiments, the polarization direction of the incident laser was rotated from 0° to 360° step-by-step with a step length of 10°.

## 3. Results and Discussions

Figure 2a shows the crystalline structure of black phosphorus, where the x-axis is along the armchair (AC) crystalline orientation, the y-axis is perpendicular to the black phosphorus plane, and the z-axis is along the zigzag (ZZ) crystalline orientation. Black phosphorus has six Raman active vibration modes. In the back-scattering geometry configuration, only the A_g_ (including A_g_^1^and A_g_^2^) and B_2g_ modes are Raman-visible [35], where A_g_^1^, A_g_^2^, and B_2g_ correspond to the atoms vibrations along the out-of-plane, AC, and ZZ direction, respectively, as shown in Figure 2b. Table 1 shows the Raman tensors of the active modes, where *a*, *b*, *c*, *h*, *f* and *g* are tensor elements.

Figure 3a is an optical image of a phosphorene sample. The cross point in the dashed box is the position of the phosphorene measured by micro-Raman spectroscopy. AC and ZZ represent the crystalline orientation of the phosphorene, *φ* is the polarization angle of the incident laser relative to the horizontal X-axis, and *θ* is the angle of the ZZ direction (z-axis in Figure 2a) with respect to the X-axis. Figure 3b is a Raman spectrum of the phosphorene in the PP and NA configurations, respectively, when *φ* = 0°. The peak positions of A_g_^1^, B_2g_ and A_g_^2^ modes correspond to about 364 cm^−1^, 442 cm^−1^ and 471 cm^−1^, respectively [36].

The layer number of the phosphorene sample can be identified by the Raman spectroscopy [37,38], atom-force microscopy (AFM) [8], photoluminescence spectroscopy [22], or optical contrast [36]. In this work, the layer numbers were determined by using the color of the phosphorene sample under the optical microscope [39] and the peak position difference between A_g_^2^ and B_2g_ modes in the Raman spectrum [36]. The peak position difference between A_g_^2^ and B_2g_ modes was 29.4 cm^−1^, which means the phosphorene sample shown in Figure 3a was three layers.

Figure 3c,d give the normalized Raman intensity (300–500 cm^−1^) with the polarization direction change of the incident laser from 0° to 360° under the PP and NA configurations, respectively. Based on the experimental results shown in Figure 3c,d, the peak intensities of A_g_^1^, B_2g_ and A_g_^2^ modes in polar coordinates are given in Figure 4, where Figure 4a–d are the polar plots under the PP and NA configurations, respectively.

Based on Table 1, the Raman intensity or each mode can be expressed as Equation (1) [35]
(1)$$I_{s}\propto {\left|e_{i}\cdot R\cdot e_{s}^{T}\right|}^{2}$$
where *e_i_* and *e_s_* are the unit vectors of the incident laser arrived to the sample and the scattering light finally detected by the spectrometer, respectively, and *R* denotes the Raman tensor. According to Equation (1), if the Raman intensity of B_2g_ mode under the NA configuration is *f*^2^, it should not change with the polarization direction of the incident laser. However, the actual data shown in Figure 4c are scarcely in a circular configuration. This is because the absorption is considered here.

The influence of anisotropic absorption on the Raman tensor needs to be considered when calculating the Raman intensity of multi-layer black phosphorus [5,33,40,41]. The Raman tensor elements *a*, *b*, *c*, *f* in Table 1 can be expressed as Equation (2) [5,35]
(2)$$a=\left|a\right|e^{i\phi_{a}},\;b=\left|b\right|e^{i\phi_{b}},\;c=\left|c\right|e^{i\phi_{c}},\;f=\left|f\right|e^{i\phi_{f}},$$
where the phases of Raman tensor elements *a*, *b*, *c*, *f* are
(3)$$\phi_{a}=\arctan\left[\frac{\frac{\partial \varepsilon_{xx}^{\prime \prime }}{\partial q^{A_{g}}}}{\frac{\partial \varepsilon_{xx}^{\prime }}{\partial q^{A_{g}}}}\right],\;\phi_{b}=\arctan\left[\frac{\frac{\partial \varepsilon_{yy}^{\prime \prime }}{\partial q^{A_{g}}}}{\frac{\partial \varepsilon_{yy}^{\prime }}{\partial q^{A_{g}}}}\right],\;\phi_{c}=\arctan\left[\frac{\frac{\partial \varepsilon_{zz}^{\prime \prime }}{\partial q^{A_{g}}}}{\frac{\partial \varepsilon_{zz}^{\prime }}{\partial q^{A_{g}}}}\right],\;\phi_{f}=\arctan\left[\frac{\frac{\partial \varepsilon_{xz}^{\prime \prime }}{\partial q^{B_{2g}}}}{\frac{\partial \varepsilon_{xz}^{\prime }}{\partial q^{B_{2g}}}}\right],$$
respectively, *ε′_ii_* and *ε″_ii_* (*i* = *x*, *y*, *z*) represent the real part and the imaginary part of the dielectric function *ε_ij_* along different crystalline directions, *ε_ij_* = *ε’_ij_* + *iε″_ij_*, and *q^Ag^*, *q^B2g^* are the normal coordinates of A_g_ Raman mode and B_2g_ mode, respectively.

In this work, we denoted the absorption rates of incident laser at each layer along the AC and ZZ directions to be Γ and Λ, respectively, and those of scattering light along the AC and ZZ directions to be ζ and χ, respectively. The total absorption of the incident laser and scattering light along the AC, ZZ were denoted as A, B and M, N, respectively. Hence,
(4)$$A=\int_{0}^{d}\Gamma \cdot {\left(1-\Gamma \right)}^{\frac{t}{0.53}}{}^{-1}dt=\int_{0}^{d}\Gamma \cdot {\left(1-\Gamma \right)}^{1.89t-1}dt=\Gamma \cdot \frac{{\left(1-\Gamma \right)}^{1.89d-1}-{\left(1-\Gamma \right)}^{-1}}{1.89\ln\left(1-\Gamma \right)}$$
(5)$$B=\int_{0}^{d}\Lambda \cdot {\left(1-\Lambda \right)}^{\frac{t}{0.53}-1}dt=\int_{0}^{d}\Lambda \cdot {\left(1-\Lambda \right)}^{1.89t-1}dt=\Lambda \cdot \frac{{\left(1-\Lambda \right)}^{1.89d-1}-{\left(1-\Lambda \right)}^{-1}}{1.89\ln\left(1-\Lambda \right)}$$
(6)$$M=\int_{0}^{d}\zeta \cdot {\left(1-\zeta \right)}^{\frac{t}{0.53}}{}^{-1}dt=\int_{0}^{d}\zeta \cdot {\left(1-\zeta \right)}^{1.89t-1}dt=\zeta \cdot \frac{{\left(1-\zeta \right)}^{1.89d-1}-{\left(1-\zeta \right)}^{-1}}{1.89\ln\left(1-\zeta \right)}$$
(7)$$N=\int_{0}^{d}\chi \cdot {\left(1-\chi \right)}^{\frac{t}{0.53}}{}^{-1}dt=\int_{0}^{d}\chi \cdot {\left(1-\chi \right)}^{1.89t-1}dt=\chi \cdot \frac{{\left(1-\chi \right)}^{1.89d-1}-{\left(1-\chi \right)}^{-1}}{1.89\ln\left(1-\chi \right)}$$
where *d* is the thickness of black phosphorus.

According to the X-Y sample coordinate system of phosphorene shown in Figure 3a, the unit vector of the incident laser relative to the crystalline axis is *e_i_* = [sin(*φ* − *θ*), 0, cos(*φ* − *θ*)]. Therefore, the Raman intensity of B_2g_ mode under the PP configuration (*e_i_* = *e_s_*) is given by Equation (8).
(8)$$\begin{array}{ll} I_{B_{2g}}^{∥} & ={\left|e_{i}\cdot R\cdot e_{s}^{T}\right|}^{2}+K^{2}\\ & ={\left|\begin{array}{l} {\left[1-A\sin^{2}\left(\varphi -\theta \right)-B\cos^{2}\left(\varphi -\theta \right)\right]}^{\frac{1}{2}}\left[\begin{array}{ccc} \sin\left(\varphi -\theta \right) & 0 & \cos\left(\varphi -\theta \right) \end{array}\right]\\ \left[\begin{array}{ccc} 0 & 0 & \left|f\right|e^{i\phi_{f}}\\ 0 & 0 & 0\\ \left|f\right|e^{i\phi_{f}} & 0 & 0 \end{array}\right]\left[\begin{array}{c} \sin\left(\varphi -\theta \right)\\ 0\\ \cos\left(\varphi -\theta \right) \end{array}\right]{\left[1-M\sin^{2}\left(\varphi -\theta \right)-N\cos^{2}\left(\varphi -\theta \right)\right]}^{\frac{1}{2}} \end{array}\right|}^{2}+K^{2}\\ & =\left[1-A\sin^{2}\left(\varphi -\theta \right)-B\cos^{2}\left(\varphi -\theta \right)\right]\cdot \left[1-M\sin^{2}\left(\varphi -\theta \right)-N\cos^{2}\left(\varphi -\theta \right)\right]\\ & \cdot {\left|f\right|}^{2}{\left[2\sin\left(\varphi -\theta \right)\cos\left(\varphi -\theta \right)\right]}^{2}+K^{2} \end{array}$$
where K is the constant denoting the effect of incomplete linear polarization of the incident laser [42]. Similarly, under the orthogonal polarization configuration (OP), the unit vector of scattering light with respect to the crystalline axis is *e_s_* = [cos (*φ* − *θ*), 0, −sin (*φ* − *θ*)]. Therefore, the Raman intensity of B_2g_ mode under the OP configuration is given by Equation (9).

(9)$$\begin{array}{ll} I_{B_{2g}}^{⊥} & ={\left|e_{i}\cdot R\cdot e_{s}^{T}\right|}^{2}+K^{2}\\ & ={\left|\begin{array}{l} {\left[1-A\sin^{2}\left(\varphi -\theta \right)-B\cos^{2}\left(\varphi -\theta \right)\right]}^{\frac{1}{2}}\left[\begin{array}{ccc} \sin\left(\varphi -\theta \right) & 0 & \cos\left(\varphi -\theta \right) \end{array}\right]\\ \left[\begin{array}{ccc} 0 & 0 & \left|f\right|e^{i\phi_{f}}\\ 0 & 0 & 0\\ \left|f\right|e^{i\phi_{f}} & 0 & 0 \end{array}\right]\left[\begin{array}{c} \cos\left(\varphi -\theta \right)\\ 0\\ -\sin\left(\varphi -\theta \right) \end{array}\right]{\left[1-M\sin^{2}\left(\varphi -\theta \right)-N\cos^{2}\left(\varphi -\theta \right)\right]}^{\frac{1}{2}} \end{array}\right|}^{2}+K^{2}\\ & =\left[1-A\sin^{2}\left(\varphi -\theta \right)-B\cos^{2}\left(\varphi -\theta \right)\right]\cdot \left[1-M\sin^{2}\left(\varphi -\theta \right)-N\cos^{2}\left(\varphi -\theta \right)\right]\\ & \cdot {\left|f\right|}^{2}{\left[\cos^{2}\left(\varphi -\theta \right)-sin^{2}\left(\varphi -\theta \right)\right]}^{2}+K^{2} \end{array}$$

Generally, the Raman data under the NA configuration are the summation of those under the PP and OP configurations [43]. Therefore, the B_2g_ mode under the NA configuration can be expressed as the summation of Equations (8) and (9) [44,45,46]
(10)$$I_{B_{2g}}=\left[1-A\sin^{2}\left(\varphi -\theta \right)-B\cos^{2}\left(\varphi -\theta \right)\right]\cdot \left[1-M\sin^{2}\left(\varphi -\theta \right)-N\cos^{2}\left(\varphi -\theta \right)\right]\cdot {\left|f\right|}^{2}+K^{2}$$

The experimental data of the B_2g_ mode under the PP and NA configurations were fitted by using Equations (8) and (10), and the fitting results are given as the *I*_B-PP_ and *I*_B-NA_ curves in Figure 4a,c, respectively. Similarly, the experimental data of the A_g_^2^ mode under the PP and NA configurations were fitted by the models given in published papers [47], and the results are given as the *I*_A2-PP_ and *I*_A2-NA_ curves in Figure 4b,d, respectively.

From Figure 4, it can be seen that the shapes of the *I*_A2-PP_ and *I*_A2-NA_ curves all appear as waist-cocoons, each of which has its extremum (denoted as *I*_EL_) and extremum direction (denoted as *φ*_EL_). A number of published studies proved that the extremum direction *φ*_EL_, viz. the length direction, of *I*_A2-PP_ curve corresponds to the AC direction. Its orthogonal direction is the ZZ direction *θ*. However, some previous studies showed that, owing to the influence of light absorption and the interference enhancement [5,32,35], the Raman intensity in the ZZ direction under the PP configuration may increase with the phosphorene layer. Hence, the *I*_A2-PP_ shape of multi-layer phosphorene always appears as a superposition of two orthogonal spindles, generally one long and one short. Sometimes, the ZZ direction *θ* is parallel to the long spindle but sometimes parallel to the short one, which often leads to the misidentification of crystalline orientation.

The shape of the *I*_B-PP_ curve appears as the superposition of two orthogonal Arab number “8”s. In Figure 4a, the two “8”s have the same shape and height, which is identical to what Equation (9) describes. Neither of the two perpendicular extremum directions define the AC direction or ZZ direction and vice versa. Hence, the B_2g_ mode under the PP configuration cannot be used to identify the crystalline orientation of the black phosphorus.

In Figure 4c, the shape of the *I*_B-NA_ curve seems like the superposition of two orthogonal spindles, generally one long and one short. There are two extrema in this *I*_B-NA_ curve, corresponding to the vertexes of the two orthogonal spindles. The two extrema are denoted as *I*_EL_ and *I*_ES_ (*I*_EL_ ≥ *I*_ES_) and their corresponding polarization directions are *φ*_EL_ and *φ*_ES_ (*φ*_EL_ = *φ*_ES_ ± 90°), respectively.

The existing methods for identifying the crystalline orientation by Raman spectroscopy are mostly implemented in the PP configuration through A_g_^2^ mode. Previous studies have pointed out that the A_g_^2^ mode is affected by the thicknesses of black phosphorus when identifying crystalline orientations. Our previous work [47] identified the crystalline orientation through A_g_^1^ and A_g_^2^ modes under the NA configuration.

The new identification method of crystalline orientation proposed in this work is as follows. The angle-resolved Raman data can be detected through any universal micro-Raman spectroscope without an analyzer. The *I*_B-NA_ curve is fitted using Equation (10) based on the experimental data. The crystalline orientation is roughly recognizable from the fitted curve, since the ZZ direction *θ = φ*_E__L_
*= φ*_ES_ ± 90°. The accurate value of *θ* should be confirmed by using the fitted parameters. If there is bo accessory to control the polarization direction of the incident laser, rotating the sample is an alternative. Compared with PP polarization, the NA configuration has lower requirements on the conditions of Raman system and has the advantages of wider applications.

In Figure 4b, *φ*_EL_ of A_g_^2^ mode under the PP is 106.7°, which means the ZZ direction *θ =* 106.7° − 90° = 16.7°. In Figure 4c, *φ*_EL_ of the B_2g_ mode under the NA is 21.2°, therefore *θ =* 21.2°. The identification result using the traditional method is close to that using the method proposed in this work, which indicated that the B_2g_ mode under the NA configuration can be used to identify the crystalline orientation.

Figure 5a shows an optical image of the multi-layer sample used in this work. The Raman data were detected from four sampling spots on this sample, denoted as a_1_ to a_4_. From the image, the colors of the sample at different sampling spots are quite dissimilar, which means that the sampling spots had different thicknesses or layer numbers when they were detected by micro-Raman spectroscopy. At each sampling spot, the Raman spectra were recorded in different polarization directions, denoted as *φ*, of the incident laser arriving at the sample under the NA configuration. Figure 5b gives the Raman spectra detected from a_1_ to a_4_ under the NA when *φ =* 0°.

From Figure 5b, the A_g_^1^, B_2g_, and A_g_^2^ modes correspond to the Raman peaks at approximately 363 cm^−1^, 439 cm^−1^, and 467 cm^−1^, respectively. According to all experimental data, the distance between the A_g_^2^ and B_2g_ modes was approximately 27.7 cm^−1^ at most, which means that both of the black phosphorus samples were larger than five layers, belonging to multi-layer phosphorene, when they were measured in this work [36].

Figure 6 shows the B_2g_ intensity data under the NA configuration (left) and A_g_^2^ intensity data under the PP configuration (right) of all sampling spots, where the square points are the experimental data and solid lines are *I*_B-NA_(*φ*, *θ*) and *I*_A2-PP_ curves fitted using Equation (10) and Equation [47], respectively, based on the experimental data of all sampling spots. Table 2 lists the identification results of the ZZ directions for all the sampling spots using the traditional method (based on the A_g_^2^ mode under the PP configuration) and the proposed method (based on the B_2g_ mode under the NA configuration).

As mentioned above and shown in Figure 5, the sample is a piece mechanically stripped from a black phosphorus crystal. Hence, all the sampling spots should have the same ZZ direction, even though they have different layers. The results in Figure 6 and Table 2 show that the ZZ direction identified by using the B_2g_ data under the NA configuration are similar with one another, but those using the A_g_^2^ data under the PP configuration are hardly identical. A total of three out of four are misidentified to the vertical orientation of the ZZ direction. In Figure 6, the *I*_EL_ angles, *φ*_EL_, of the B_2g_ mode are consistent in different thickness regions, indicating that the results of crystalline orientation identification are consistent. Compared with the B_2g_ mode, the *φ*_EL_ of the A_g_^2^ mode from a_1_ to a_4_ change between armchair and zigzag directions depending on the sample thickness, and the results of identification of A_g_^2^ mode are contradictory. The interference enhancement factor is a function of the sample thickness for ZZ and AC directions.

In Figure 6, the results of identification of the two methods are inconsistent at the sampling spots a_1_ to a_3_, but at the a_4_, the results of identification of the two methods are consistent. It proved that the existing A_g_^2^ mode method showed a misidentification when identifying the crystalline orientation.

This is induced by the influence of the absorption, the partial depolarization, and the interference enhancement effect [5,32,35], each of which has its respective influencing mechanism and is strongly linked to the sample thickness and the polarization direction of either the incident laser or the scattering light because of the anisotropy and layer number change of the multilayer phosphorene sample. In the PP configuration, the polarization direction arriving at the sample is parallel to that of the scattering light detected by the spectrometer. Hence, the influence of absorption, depolarization, and interference on the angle-resolved Raman intensity in each polarization direction is superimposed and found to be more sensitive to the number of layers. By contrast, when under the NA configuration, the scattering data in all the polarization directions are collected by the spectrometer, which weakens the influences of partial depolarization and the interference enhancement effect on the Raman scattering signal with angle-resolved polarization to a certain extent. Black phosphorus is an anisotropic material. Due to the linear dichroism of black phosphorus and the effect of light absorption, the polar graph of the A_g_^2^ mode changed from peanut to a superposition of two orthogonal spindles. Because of its dichroism and birefringence, black phosphorus has a polarization dependent interference enhancement. The interference enhancement factors of ZZ and AC direction are different, and when the polarization direction of the incident laser is along ZZ direction, the Raman scattering intensity can be enhanced much more than that of the AC direction.

## 4. Conclusions

This paper presented an experimental and theoretical analysis of the identification of the crystalline orientation for multilayer phosphorene samples based on the B_2g_ mode under the nonanalyzer configuration. The identification method was proposed based on the model of the angle-resolved Raman intensity influenced by the absorption, the partial depolarization, and the interference enhancement effect. Compared with the results obtained using the traditional method, which applies the A_g_^2^ mode under a parallel configuration, the proposed method showed a higher effectivity free from misidentification, as well as higher fault tolerance for different thicknesses. The model established in this paper can accurately fit the values of crystalline orientation. The proposed method can be used to identify the crystalline orientation of few and multi-layer black phosphorus. This method reduces the requirements of the Raman system and has a wider application range. The method is beneficial to the study of the Raman active mode and anisotropic nature of black phosphorus. Furthermore, this method provides great help for studying the anisotropic characteristics of electron, phonon and interaction of black phosphorus.

## Figures and Tables

**Figure 1 materials-13-05572-f001:**
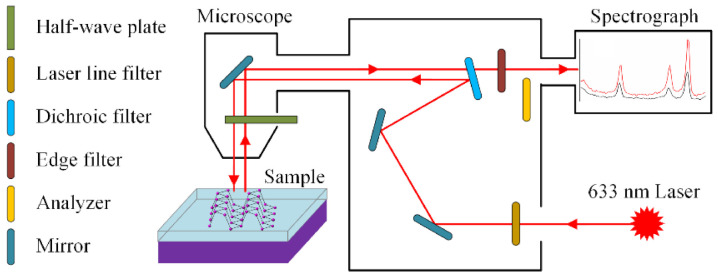
Diagram of polarized micro-Raman spectroscope.

**Figure 2 materials-13-05572-f002:**
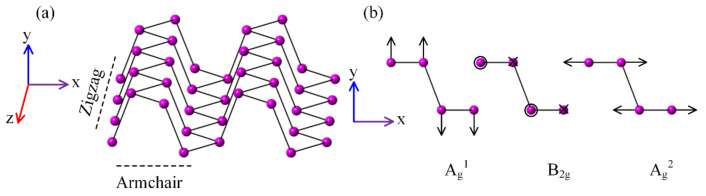
(**a**) Crystalline structure of black phosphorus, (**b**) atomic displacements of the Raman-active modes in black phosphorus.

**Figure 3 materials-13-05572-f003:**
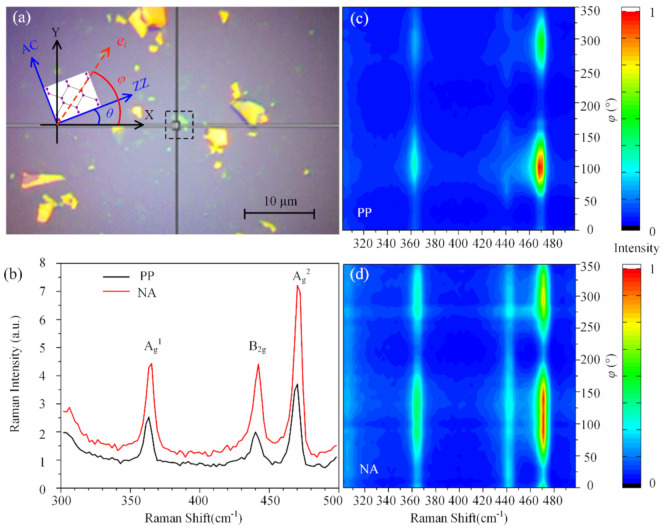
(**a**) Optical microscope image of the three-layers black phosphorus sample, (**b**) Raman spectrum when the polarization angle of incident laser at 0° under the parallel polarization (PP) and nonanalyzer (NA) configurations, (**c**,**d**) normalized Raman intensity (300–500 cm^−1^) with the polarization direction change of the incident laser from 0° to 360° under the PP and NA configurations, respectively.

**Figure 4 materials-13-05572-f004:**
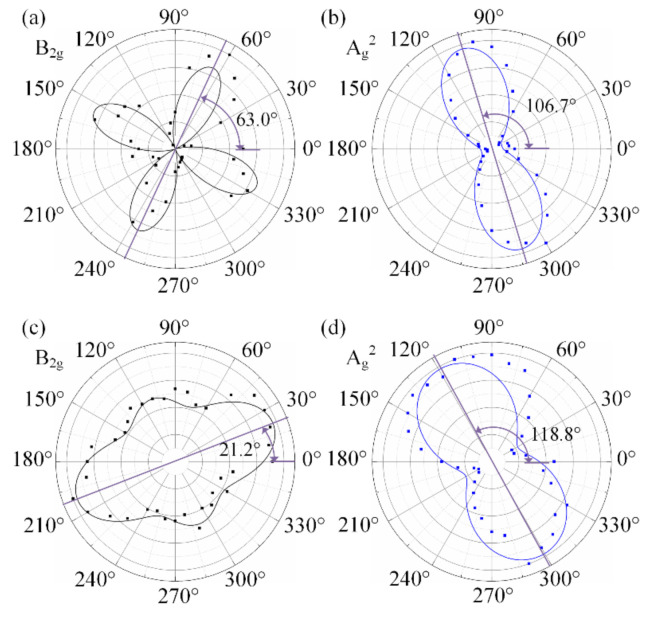
Polar plots of normalized intensities of (**a**) B_2g_ mode under the PP configuration, (**b**) A_g_^2^ mode under the PP configuration, (**c**) B_2g_ mode under the NA configuration, (**d**) A_g_^2^ mode under the NA configuration, respectively.

**Figure 5 materials-13-05572-f005:**
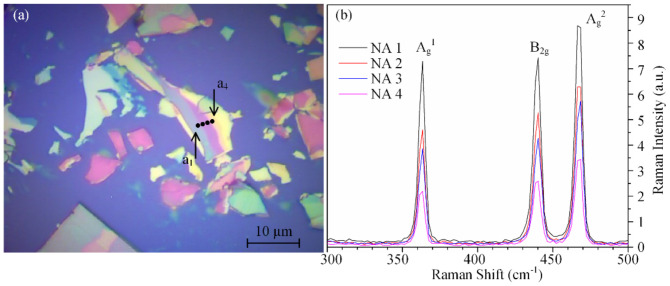
(**a**) Optical image of black phosphorus sample, where a_1_ to a_4_ are the polarized micro-Raman sampling spots, (**b**) Raman spectra detected from a_1_ to a_4_ under the NA configuration when *φ* = 0°.

**Figure 6 materials-13-05572-f006:**
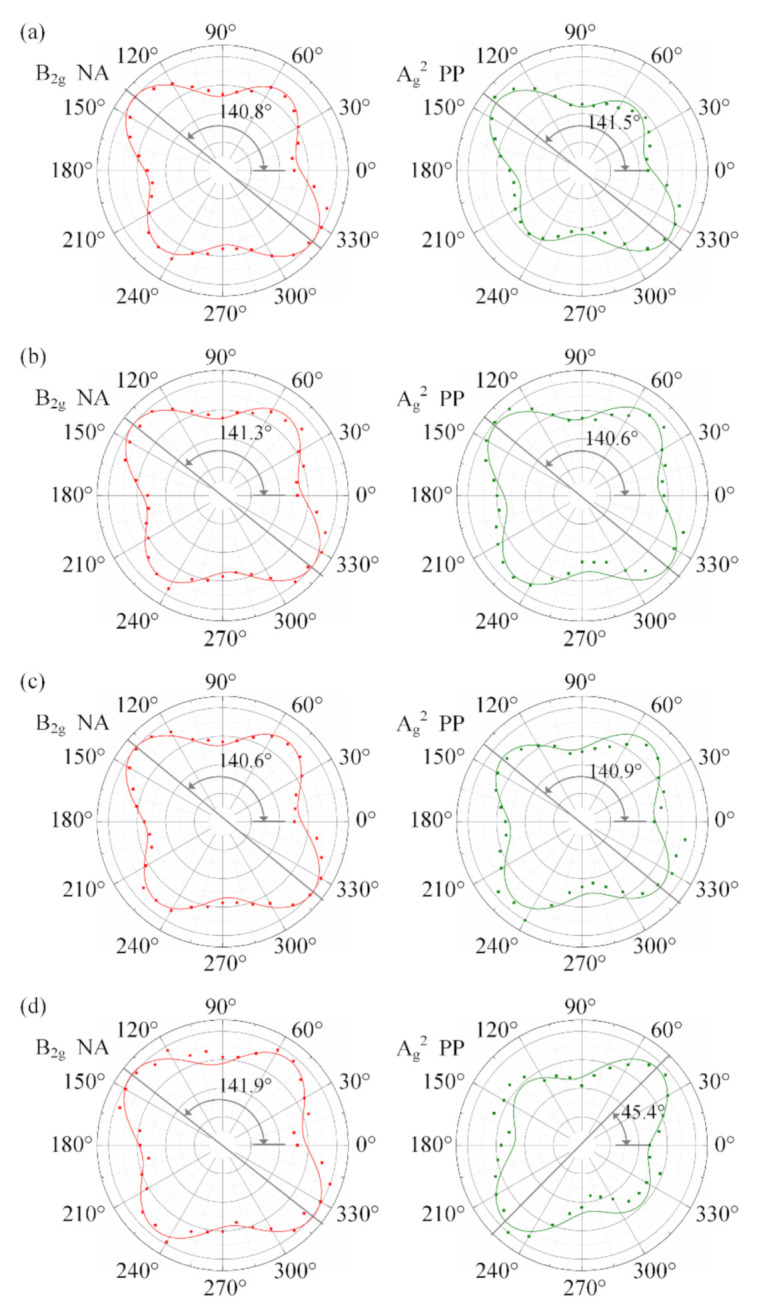
Polar plots of normalized intensities of sampling spots (**a**) a_1_, (**b**) a_2_, (**c**) a_3_, and (**d**) a_4_, where the left are the B_2g_ data under the NA configuration and the right are A_g_^2^ data under the PP configuration.

**Table 1 materials-13-05572-t001:** Raman tensors of all active modes of black phosphorus.

A_g_	B_1g_	B_2g_	B_3g_
$R\left(A_{g}\right)=\left(\begin{array}{ccc} a & 0 & 0\\ 0 & b & 0\\ 0 & 0 & c \end{array}\right)$	$R\left(B_{1g}\right)=\left(\begin{array}{ccc} 0 & h & 0\\ h & 0 & 0\\ 0 & 0 & 0 \end{array}\right)$	$R\left(B_{2g}\right)=\left(\begin{array}{ccc} 0 & 0 & f\\ 0 & 0 & 0\\ f & 0 & 0 \end{array}\right)$	$R\left(B_{3g}\right)=\left(\begin{array}{ccc} 0 & 0 & 0\\ 0 & 0 & g\\ 0 & g & 0 \end{array}\right)$

**Table 2 materials-13-05572-t002:** The identification results of the armchair (AC) directions for all the sampling spots obtained using the traditional method (based on the A_g_^2^ mode under the PP configuration) and the proposed method (based on the B_2g_ mode under the NA configuration).

Sampling Spot		a_1_	a_2_	a_3_	a_4_
B_2g_	NA	140.8°	141.3°	140.6°	141.9°
A_g_^2^	PP	141.5° − 90° = 51.5°	140.6° − 90° = 50.6°	140.9° − 90° = 50.9°	45.4° + 90° = 135.4°

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
