# Peer review of "Orientation Identification of the Black Phosphorus with Different Thickness Based on B2g Mode Using a Micro-Raman Spectroscope under a Nonanalyzer Configuration"

_materials, 2020, doi:10.3390/ma13235572_

Round 1

Reviewer 1 Report

In this paper, the authors introduced a new method in this paper to identify the crystalline orientation of black phosphorus. This method is based on the angle-resolved Raman detection and the theoretical model of intensity-orientation relationship of the B2g mode under the nonanalyzer configuration. The idea behind this is interesting. However, I still have quite a number of concerns in this manuscript. There are times where there are not enough data to support the conclusions of the author. Please see some of the major concerns below.

1.The information for Diagram of polarized micro-Raman spectroscope is not enough. The authors should give much more information about this. So the readers can get its reproducibility.  For example what is the resolution of the spectroscope? more details for the optical path and etc.

  1. The authors should give much more information about the novelty of this paper, especially the effect of using this new method for Raman detection, which applications can be used this method?

  1. More references need to be included in the introduction part to understand the applications of using Raman detection system by improving resolution:

Improving Raman spectra of pure silicon using super-resolved method

- Journal of Optics, 2019

Super-resolved Raman spectra of toluene and toluene–chlorobenzene mixture

- Spectroscopy Letters, 2015

  1. Much more discussion about the results should be given in this paper, especially the author needs to provide enough physicals mechanism analysis about the results.

Author Response

Question 1:

The information for Diagram of polarized micro-Raman spectroscope is not enough. The authors should give much more information about this. So the readers can get its reproducibility. For example what is the resolution of the spectroscope? more details for the optical path and etc.

Answer:

Thanks for the reviewer’s instructive suggestions. According to the reviewer’s key suggestion, we have supplemented the required information in Section 2.2 (Paragraph 5, Page 2), which was marked in red in the revised manuscript.

Question 2:

The authors should give much more information about the novelty of this paper, especially the effect of using this new method for Raman detection, which applications can be used this method?

Answer:

According to the reviewer’s suggestion, we have given more information about the new method and applications in Section 1 (Paragraph 3, Page 2), which was marked in red in the revised manuscript.

Question 3:

More references need to be included in the introduction part to understand the applications of using Raman detection system by improving resolution:

Improving Raman spectra of pure silicon using super-resolved method

- Journal of Optics, 2019

Super-resolved Raman spectra of toluene and toluene–chlorobenzene mixture

- Spectroscopy Letters, 2015

Answer:

According to the reviewer’s suggestion, we add the recommended references in Section 1 (Paragraph 1, Page 2), which was marked in red in the revised manuscript.

[27] Malka, D.; Adler Berke, B.; Tischler, Y.; Zalevsky, Z. Improving Raman spectra of pure silicon using super-resolved method. J. Optics-UK 2019, 21, 075801.

[28] Malka, D.; Berkovic, G.; Tischler, Y.; Zalevsky, Z. Super-resolved Raman spectra of toluene and toluene–chlorobenzene mixture. Spectrosc. Lett. 2015, 48, 431-435.

Question 4:

Much more discussion about the results should be given in this paper, especially the author needs to provide enough physicals mechanism analysis about the results.

Answer:

According to the reviewer’s suggestion, we discuss the results of this paper more and give a more adequate physical mechanism analysis in Section 3 (Paragraph 1, Page 9), which was marked in red in the revised manuscript.

Reviewer 2 Report

In paper, a newly-developed method of the identification of the crystalline orientation of the black phosphorus has been presented. The main concept of the article seems to be interesting and meaningful due to the fact that such an orientation determines the properties of the mentioned phosphorus which, in turn, is essential in viewpoint of its application. In general, article has been prepared properly but some corrections are suggested – all comments are presented in more detail below.

  • Abstract of the paper: more attention should be paid to the most important results of the investigations performed.
  • Introduction: Authors should provide more information on the unique characteristics of phosphorene referring to the adequate literature reports.
  • During the analysis of the results obtained Authors should extend the discussion over the differences between the existing methods of the identification of the crystalline orientation of the phosphorus and the method proposed in this paper.
  • The issue of the impact of the absorption, depolarization and the interference enhancement effect on the angle-resolved Raman intensity should be more widely discussed in the last paragraph of the section 3.
  • Conclusions of the paper are too general. In this section Authors should emphasize the highlights of the research and the importance of the conducted studies.

Author Response

Question 1:

Abstract of the paper: more attention should be paid to the most important results of the investigations performed.

Answer:

Thanks for the reviewer’s instructive suggestions. In this paper, a crystalline orientation identification model of B2g mode is also established, and the method proposed in this paper can identify black phosphorus at different layers. We have modified the abstract of this article, and the most important results have been described more. We give more attention about the abstract of the paper (Page 1), which was marked in red in the revised manuscript.

Question 2:

Introduction: Authors should provide more information on the unique characteristics of phosphorene referring to the adequate literature reports.

Answer:

According to the reviewer’s suggestion, we add the appropriate references in the introduction (Paragraph 1, Page 1). It is necessary to supplement more references on the characteristics of black phosphorus in this paper, so that readers can directly understand the application field and value of black phosphorus. The relevant sentences were marked in red in the revised manuscript.

[11] Rodin, A. S.; Carvalho, A.; Castro Neto, A. H. Strain-induced gap modification in black phosphorus. Phys. Rev. Lett. 2014, 112, 176801.

[12] Tran, V.; Soklaski, R.; Liang, Y.; Yang, L. Layer-controlled band gap and anisotropic excitons in few-layer black phosphorus. Phys. Rev. B 2014, 89, 235319.

[13] Na, J.; Lee, Y. T.; Lim, J. A.; Hwang, D. K.; Kim, G.-T.; Choi, W. K.; Song, Y.-W. Few-layer black phosphorus field-effect transistors with reduced current fluctuation. ACS Nano 2014, 8, 11753-11762.

[14] Liu, H.; Neal, A. T.; Zhu, Z.; Luo, Z.; Xu, X.; Tomanek, D.; Ye, P. D. Phosphorene: an unexplored 2D semiconductor with a high hole mobility. ACS Nano 2014, 8, 4033-4041.

[15] Du, Y.; Maassen, J.; Wu, W.; Luo, Z.; Xu, X.; Ye, P. D. Auxetic black phosphorus: a 2D material with negative poisson’s ratio. Nano Lett. 2016, 16, 6701-6708.

[16] Jiang, J.-W.; Park, H. S. Negative poisson’s ratio in single-layer black phosphorus. Nat. Commun. 2014, 5, 4727.

[17] Qiao, J.; Kong, X.; Hu, Z.-X.; Yang, F.; Ji, W. High-mobility transport anisotropy and linear dichroism in few-layer black phosphorus. Nat. Commun. 2014, 5, 4475.

[18] Zhang, G.; Huang, S.; Chaves, A.; Song, C.; Özçelik, V. O.; Low, T.; Yan, H. Infrared fingerprints of few-layer black phosphorus. Nat. Commun. 2017, 8, 14071.

[19] Mak, K. F.; Lee, C.; Hone, J.; Shan, J.; Heinz, T. F. Atomically thin MoS2: a new direct-gap semiconductor. Phys. Rev. Lett. 2010, 105, 136805.

[20] Ling, X.; Wang, H.; Huang, S.; Xia, F.; Dresselhaus, M. S. The renaissance of black phosphorus. Proc. Natl. Acad. Sci. USA 2015, 112, 4523-4530.

Question 3:

During the analysis of the results obtained Authors should extend the discussion over the differences between the existing methods of the identification of the crystalline orientation of the phosphorus and the method proposed in this paper.

Answer:

The existing method recognizes the crystalline orientation of black phosphorus through Ag2 mode, while the proposed method recognizes the crystalline orientation through B2g mode. In the experiment of this paper, the existing method has misidentification of crystalline orientation, but the method in this paper does not. According to the reviewer’s suggestion, we give more discussions over the differences between the existing methods and the method proposed in Section 3 (paragraph 2, Page 9), which was marked in red in the revised manuscript.

Question 4:

The issue of the impact of the absorption, depolarization and the interference enhancement effect on the angle-resolved Raman intensity should be more widely discussed in the last paragraph of the section 3.

Answer:

Light absorption, depolarization and interference enhancement will affect the Raman intensity of black phosphorus B2g mode, which may lead to misidentification of crystalline orientation of black phosphorus. According to the reviewer’s suggestion, we give more discussions about the impact of the absorption, depolarization and the interference enhancement effect in Section 3 (paragraph 3, Page 9), which was marked in red in the revised manuscript.

Question 5:

Conclusions of the paper are too general. In this section Authors should emphasize the highlights of the research and the importance of the conducted studies.

Answer:

According to the reviewer’s suggestion, the highlights of this paper and the importance of the research are further described to help clarify and emphasize the key points of this paper in Section 4 (Page 10), which was marked in red in the revised manuscript.

Reviewer 3 Report

  The authors describe very interesting results, but there are still some points that need to be addressed:   For the 1. introduction: The following review article from 2017 should be cited and its central statements in the introduction should be classified in a technical way: https://doi.org/10.1002/jrs.5238   Unfortunately only very few applications are described. However, 2-3 current publications should be cited.   To the experimental part: All experiments and equipment used should be described, for example the light microscope would not be used and what software was used for evaluation?    To Results and Discussion:
See Fig. 3: The spectra should be compared with literature spectra and the literature should be cited. The resolution of the microscope images (see figures 3 and 5) is not very good. Please provide images with better resolution.
  To 4. Conclusion: The overall conclusion is too brief. What is the greater effectiveness and how can we classify the tolerance for error? In addition, the literature is not discussed and no direct comparison is made at this point.   Nothing more is mentioned about the outlook. It would be good if we could go into this briefly here.    

Author Response

Question 1:

For the 1. introduction: The following review article from 2017 should be cited and its central statements in the introduction should be classified in a technical way: https://doi.org/10.1002/jrs.5238 Unfortunately only very few applications are described. However, 2-3 current publications should be cited.

Answer:

Thanks for the reviewer’s instructive suggestions. References recommended by reviewers and those related to the application of black phosphorus have also been added to the introduction, and it is necessary to add these references related to the application of black phosphorus. We add the appropriate references in the introduction (Paragraph 1, Page 1 and Page 2), and the relevant sentences were marked in red in the revised manuscript.

[1] Li, L.; Yu, Y.; Ye, G. J.; Ge, Q.; Ou, X.; Wu, H.; Feng, D.; Chen, X. H.; Zhang, Y. Black phosphorus field-effect transistors. Nat. Nanotechnol. 2014, 9, 372-377.

[8] Xia, F.; Wang, H.; Jia, Y. Rediscovering black phosphorus as an anisotropic layered material for optoelectronics and electronics. Nat. Commun. 2014, 5, 4458.

[12] Tran, V.; Soklaski, R.; Liang, Y.; Yang, L. Layer-controlled band gap and anisotropic excitons in few-layer black phosphorus. Phys. Rev. B 2014, 89, 235319.

[14] Liu, H.; Neal, A. T.; Zhu, Z.; Luo, Z.; Xu, X.; Tomanek, D.; Ye, P. D. Phosphorene: an unexplored 2D semiconductor with a high hole mobility. ACS Nano 2014, 8, 4033-4041.

[17] Qiao, J.; Kong, X.; Hu, Z.-X.; Yang, F.; Ji, W. High-mobility transport anisotropy and linear dichroism in few-layer black phosphorus. Nat. Commun. 2014, 5, 4475.

[20] Ling, X.; Wang, H.; Huang, S.; Xia, F.; Dresselhaus, M. S. The renaissance of black phosphorus. Proc. Natl. Acad. Sci. USA 2015, 112, 4523-4530.

[21] Wu, J.; Mao, N.; Xie, L.; Xu, H.; Zhang, J. Identifying the crystalline orientation of black phosphorus using angle-resolved polarized Raman spectroscopy. Angew. Chem. Int. Ed. 2015, 54, 2366-2369.

[26] Ribeiro, H. B.; Pimenta, M. A.; de Matos, C. J. S. Raman spectroscopy in black phosphorus. J. Raman Spectrosc. 2018, 49, 76-90.

Question 2:

To the experimental part: All experiments and equipment used should be described, for example the light microscope would not be used and what software was used for evaluation?

Answer:

According to the reviewer’s suggestion, we have added descriptions of experiments and equipment, and this information will be helpful for other readers to refer to and repeat experiments in Section 2.2 (Page 2), which was marked in red in the revised manuscript.

Question 3:

To Results and Discussion: See Fig. 3: The spectra should be compared with literature spectra and the literature should be cited. The resolution of the microscope images (see figures 3 and 5) is not very good. Please provide images with better resolution.

Answer:

According to the reviewer’s suggestion, we have added the references needed to compare the experimental spectra in this paper with those in the literature in Section 3 (Paragraph 1, Page 4), which was marked in red in the revised manuscript. The low resolution of the microscope images in figures 3 and 5 is due to the low resolution of the charge coupled device (CCD) used. Therefore, we are sorry that the resolution of the optical image cannot be improved.

[36] Lu, W.; Nan, H.; Hong, J.; Chen, Y.; Zhu, C.; Liang, Z.; Ma, X.; Ni, Z.; Jin, C.; Zhang, Z. Plasma-assisted fabrication of monolayer phosphorene and its Raman characterization. Nano Res. 2014, 7, 853-859.

Question 4:

To 4. Conclusion: The overall conclusion is too brief. What is the greater effectiveness and how can we classify the tolerance for error? In addition, the literature is not discussed and no direct comparison is made at this point. Nothing more is mentioned about the outlook. It would be good if we could go into this briefly here.

Answer:

According to the reviewer’s suggestion, we give more information about the conclusions in Section 4 (Page 10), which was marked in red in the revised manuscript. The misidentification of the crystalline orientation can be reduced by the B2g method in this paper. The existing Ag2 method has a higher proportion of misidentification in crystalline orientation recognition than the method in this paper. The error tolerance in this paper refers to the proportion of correct identification in the experiment of crystalline orientation. We also give more information about the outlook.

Round 2

Reviewer 1 Report

The modified version can be published.

Reviewer 3 Report

Dear Authors,

Thank you for your suggestions and the changes.

Herewith I accept all changes.